# A Randomized Trial in Older Adults of a Flavor-Enhanced Coconut Oil-Based Mouthwash: Clinical Safety, Antimicrobial Efficacy, and User Satisfaction

**DOI:** 10.3390/healthcare13222941

**Published:** 2025-11-17

**Authors:** Arpasiri Soodsakorn, Wantida Chaiyana, Jitjiroj Ittichaicharoen, Phenphichar Wanachantararak, Marut Wongtapin, Siriwoot Sookkhee, Darunee Owittayakul

**Affiliations:** 1Department of Advanced General Dentistry and Dental Public Health, Faculty of Dentistry, Chiang Mai University, Chiang Mai 50200, Thailand; b.arpasiri@gmail.com; 2Department of Pharmaceutical Sciences, Faculty of Pharmacy, Chiang Mai University, Chiang Mai 50200, Thailand; wantida.chaiyana@cmu.ac.th; 3Department of Oral Biology and Oral Diagnostic Sciences, Faculty of Dentistry, Chiang Mai University, Chiang Mai 50200, Thailand; jitjiroj.itti@cmu.ac.th; 4Dental Research Center, Faculty of Dentistry, Chiang Mai University, Chiang Mai 50200, Thailand; phenphichar.w@cmu.ac.th; 5Department of Microbiology, Faculty of Medicine, Chiang Mai University, Chiang Mai 50200, Thailand; marut_w@cmu.ac.th

**Keywords:** anti-bacterial agents, antifungal agents, *Candida albicans*, *Cocos nucifera*, mouthwashes, oral health, patient satisfaction, randomized controlled trial

## Abstract

**Background/Objectives**: Fungal and bacterial infections are major contributors to oral diseases in older adults. Although chlorhexidine (CHX) is widely recognized for its antimicrobial efficacy, its prolonged use is constrained by adverse effects. Virgin coconut oil (VCO) possesses antimicrobial properties; however, its high viscosity reduces acceptability. This study aimed to develop a flavor-enhanced coconut oil-based mouthwash (FCoMW) and evaluate its clinical safety, antimicrobial efficacy, and user satisfaction. **Methods**: A 14-day randomized controlled trial was conducted in older adults at the Faculty of Dentistry, Chiang Mai University, Thailand (April–July, 2024). Participants were randomly allocated by simple randomization to FCoMW, CHX, and normal saline solution (NSS). The evaluator was blinded. Assessments included burning sensation, oral mucosal alterations, changes in oral *Candida* and bacterial counts, and user satisfaction. Results: Among 51 participants (NSS: 17; CHX: 16; FCoMW: 18), FCoMW significantly reduced oral *Candida* counts by Day 14 (*p* < 0.0001), with efficacy comparable to CHX. CHX achieved greater bacterial reduction (*p* < 0.05). No adverse effects occurred with FCoMW. User satisfaction was high for scent, freshness, and overall acceptability. **Conclusions**: FCoMW is safe, well-accepted, and efficacious against oral *Candida*, offering a natural alternative to chlorhexidine for long-term use in older adults.

## 1. Introduction

Microorganisms and their metabolic by-products significantly influence the disruption of oral cavity homeostasis, thereby contributing to the development of various oral diseases. Consequently, the use of antimicrobial mouthwashes may serve as an effective adjunctive strategy to reduce microbial load and maintain oral health [1]. Among older adults, particularly those who wear dentures, oral rinsing plays a crucial role in supporting oral hygiene and preventing infections such as oral candidiasis. This opportunistic fungal infection—most commonly associated with the overgrowth of *Candida albicans*—is frequently observed in the aging population [2]. Contributing risk factors include immunocompromised states, hyposalivation, and the presence of ill-fitting or inadequately maintained dentures [3]. Notably, the prevalence of denture stomatitis among removable denture wearers has been reported to range from 20% to 67%, with the condition most frequently detected in elderly patients [4].

Chlorhexidine (CHX) is a widely used mouthwash known for its broad-spectrum antimicrobial activity and proven efficacy against oral candidiasis caused by *Candida* species. Its primary mechanism involves the inhibition of biofilm formation and microbial adhesion in the oral cavity. Its antifungal efficacy is well established, particularly at a concentration of 0.12% [1]. At low concentrations, CHX exerts a bacteriostatic effect by interacting with bacterial cell walls through its cationic charge, disrupting cell membrane integrity. At higher concentrations, it induces cellular leakage and coagulates cytoplasmic components, resulting in bactericidal activity [5,6]. Furthermore, CHX effectively reduces dental plaque accumulation, thereby contributing to the prevention of oral infections. However, CHX mouthwash is associated with several notable adverse effects, including a bitter taste, extrinsic staining of teeth, composite restorations, and oral mucosa, as well as altered taste perception (dysgeusia), mucosal irritation, and potential allergic reactions [5]. Furthermore, the emergence of microbial resistance to CHX has been linked to the upregulation of efflux pump mechanisms [7]. Given these limitations, CHX mouthwashes are generally recommended for short-term use only [8]. Alternatively, mouthwashes formulated with natural or herbal extracts, possessing antimicrobial properties and inhibiting plaque formation while exhibiting fewer side effects, present a promising alternative for oral hygiene maintenance [9,10].

Interestingly, oil pulling with virgin coconut oil (VCO) has become widely accepted due to its accessibility, mild aroma, and pleasant taste [11]. It has demonstrated antimicrobial activity against *Streptococcus mutans* and *Candida albicans* [12]. A systematic review has reported the effects of VCO pulling on dental health and oral hygiene [11]. VCO has demonstrated potential oral health benefits, including caries prevention, reductions in dental plaque, gingivitis and halitosis, as well as relief from xerostomia [13,14,15,16,17,18]. Furthermore, VCO exhibits broad-spectrum antimicrobial activity, demonstrating efficacy against both Gram-negative bacteria [19], and Gram-positive [16,20], including *Clostridioides difficile* [21], as well as antiviral [22] and antifungal activities [23]. VCO provides notable benefits, with many users reporting improved oral cleanliness, a pleasant aroma, and reduced oral malodor. Nevertheless, its use in oil pulling presents certain limitations. These include high viscosity, an oily texture, and the extended rinsing duration required for effectiveness, which may hinder consistent adherence [16].

The previous study formulated a mouthwash incorporating VCO, propylene glycol, and distilled water (DW) in a 60:30:10 ratio, respectively, to reduce viscosity. The formulation exhibited significant anticandidal activity, with no statistically significant difference compared with 100,000 units/mL of nystatin (*p* > 0.05) [24]. Furthermore, its anti-inflammatory and wound-healing properties were assessed using gingival fibroblasts, revealing comparable efficacy to that of 0.12% CHX mouthwash [25]. However, phase separation between the oil and water components in this mouthwash formula presents a significant challenge. To enhance stability, a modified formulation of coconut oil-based mouthwash (CoMW) was developed, consisting of VCO, mixed emulsifier (ME), and DW in a 70:5:25 ratio. This formulation replaces propylene glycol with two emulsifiers, polysorbate 80 (Tween 80) and sorbitan oleate (Span 80), in a 40:60 ratio, resulting in a stable and homogeneous blend [26]. In in vitro studies, CoMW effectively inhibited *C. albicans* and biofilm growth after continuous exposure for at least 5 min. The inclusion of emulsifiers reduces surface tension, enhances emulsion stability, and maintains the hydrophilic-lipophilic balance (HLB) of the water-in-oil (*w*/*o*) emulsion, thereby improving its anticandidal efficacy [27].

Building on this evidence, a flavor-enhanced coconut oil-based mouthwash (FCoMW) was developed by incorporating flavoring agents and preservatives to improve palatability and user acceptability. Therefore, the aim of this randomized controlled trial was to evaluate the clinical safety, antibacterial and anticandidal efficacy, and user satisfaction of FCoMW in comparison with 0.12% *w*/*v* CHX and normal saline solution (NSS) among older adults. The null hypothesis was that no significant differences would be observed among the three groups in terms of clinical safety, antibacterial and anticandidal efficacy, and user satisfaction. In addition, the study was designed to address the clinical relevance of FCoMW as a potential long-term alternative to conventional chemical mouthwashes for preventive oral care in elderly individuals at risk of oral candidiasis and denture-related infections.

## 2. Materials and Methods

### 2.1. Volunteers and Ethical Issues

The present study was a randomized controlled trial conducted at the Comprehensive Dental Clinic, Faculty of Dentistry, Chiang Mai University, Thailand. The study protocol was approved by the Human Experimentation Committee of the Faculty of Dentistry, Chiang Mai University, on 1 March 2024 (Document No. 8/2024), and registered at ThaiClinicalTrials.org (TCTR20240711004) on 11 July 2024. Written informed consent was obtained from all participants, who demonstrated the ability to understand and adhere to the study protocol. Eligible participants were aged over 60 years and were in good general health with well-controlled medical conditions (ASA physical status I–II). All were independent older adults with normal swallowing function. *Candida* species were detected by culture from concentrated oral rinse samples, with colony counts not exceeding 400 CFU [28].

Exclusion criteria included significant or relevant medical or oral abnormalities that could interfere with the study, such as dysphagia or untreated oral mucosal diseases; intolerance or hypersensitivity to the study interventions; current smoking or smoking cessation within the past year; and the use of topical or systemic antifungal medications within one month prior to enrollment. The inclusion and exclusion criteria are summarized in Table 1.

### 2.2. Sample Size Determination

The data reported by Uludamar et al. [29] were used to calculate the sample size using the following formula in the n4Studies Android application (version 1.4.0) [30], with a significance level of 0.05 and a statistical power of 80%. The following formula was used: n = [((z_1_ − α/2) + (z_1_ − β))^2^ × (σ_1_^2^ + (σ_2_^2^/r)]/[µ_1_ − µ_2_]^2^, where σ_1_^2^ and σ_2_^2^ are the variances in the two groups, µ_1_ − µ_2_ is the expected difference in means, and r is the allocation ratio.

The minimum required sample size was calculated to be 15 participants per group.

### 2.3. FCoMW Formulation

According to this previous study [26], the CoMW formulation was composed of the portions of VCO, ME, and DW in a ratio of 70:5:25 *w*/*v*. The ME contained two emulsifiers, polysorbate 80 (Tween 80; Batch D0013RA312, Krungthepchemi Ltd., Lad Prao, Bangkok, Thailand) and sorbitan oleate 80 (Span 80; Batch SMOML0321, Top Inno Tec Co., Ltd., Suan Luang, Bangkok, Thailand), mixed at a 40:60 ratio. The ME was combined with sterilized VCO in a glass vessel, and DW was added gradually while stirring on a magnetic plate (AM4 multiple heating magnetic stirrer, VELP Scientific, Inc., Deer Park, NY, USA) until homogeneous. The pre-emulsion was then homogenized for 5 min using a high-shear homogenizer (T25 digital ULTRA-TURRAX^®^, IKA, Staufen, Germany) to yield a stable water-in-oil system. Two food-grade flavoring agents, peppermint and orange oils (Namsiang, Bangkok, Thailand), were incorporated into the formulation to enhance its taste and aroma. One percent *w*/*v* concentrated paraben (CP) consisting of 20% *w*/*v* methylparaben and 2% *w*/*v* propylparaben dissolved in propylene glycol was incorporated into FCoMW to ensure microbiological stability.

### 2.4. Evaluation of Oral Health Parameters, Safety, and Satisfaction

Participant Allocation and Study Protocol

Subjects were recruited from existing patients who attended the Comprehensive Dentistry Clinic, Faculty of Dentistry, Chiang Mai University, Thailand. Between April and July 2024, a total of 64 participants were screened for eligibility. Eight were excluded (four did not meet the inclusion criteria and four declined to participate). The remaining 56 provided written informed consent and were randomly allocated, using a computer-generated simple randomization sequence with evaluator blinding, to three treatment groups: the experimental group receiving FCoMW (n = 19), the positive control group receiving 0.12% *w*/*v* CHX (n = 19), and the negative control group receiving NSS (n = 18). Each participant received a designated mouthwash supply and a calibrated measuring cup for twice-daily use over two weeks, following toothbrushing in the morning and evening. The FCoMW and NSS groups were instructed to rinse with 15 mL of their assigned solutions for 5 min, while the CHX group rinsed with 15 mL of 0.12% *w*/*v* CHX for 30 s, in accordance with standard CHX usage guidelines. All participants were advised to avoid eating or drinking for 30 min after rinsing and to adhere strictly to the usage instructions. Participants were provided with a daily log sheet to record mouthwash usage. To eliminate variability from oral hygiene practices, all participants were supplied with the same type of toothbrush and toothpaste for the study duration. Follow-up phone calls were conducted after one week to assess compliance and monitor for any adverse effects, such as irritation or discomfort. On Day 14, the residual volume of mouthwash was measured by study personnel to verify participant compliance with the prescribed protocol. For ethical reasons, after completing the 14-day intervention and a 21-day interval, participants in the NSS group were reassigned to receive either FCoMW or CHX for continued treatment; however, no additional data were collected from these participants for research purposes, and their outcomes were not included in the analysis.

Oral Examination of Oral Health Parameters

An intraoral examination was conducted at baseline and on Day 14 for all participants, including the number of teeth, the plaque accumulation, and the quality of dentures for those wearing removable dentures. The moisture level in the oral cavity was measured using the Oral Moisture Checker Mucus (Mucus^®^, Life Co., Ltd., Saitama, Japan). Plaque accumulation was assessed using the Silness and Löe Plaque Index [31] on six representative teeth, following the application of an erythrosine-based disclosing solution. Four surfaces of each tooth, including buccal, lingual, mesial, and distal, were evaluated and scored on a scale from 0 to 3 based on the amount of plaque observed. The scoring criteria were as follows: 0 indicated no visible plaque: 1 indicated a thin film of plaque detectable at gingival margin by gently running a probe across the surface; 2 indicated moderate accumulation of plaque visible along the gingival margin; and 3 indicated heavy plaque accumulation observed at the gingival margin and within the interdental spaces. The average score for each tooth was obtained by summing the scores of the four surfaces and dividing by four. The overall plaque index for each participant was then calculated by averaging the scores of all six teeth. The examining dentist, with over five years of experience, underwent calibration with a specialist in advanced general dentistry to ensure consistency, achieving Cohen’s Kappa coefficient (κ) values exceeding 0.80 and percent agreement greater than 80%.

Oral Safety Evaluation

Oral mucosal safety was evaluated through oral soft tissue examinations conducted before use, immediately after mouthwash application, and on Day 14, along with any adverse events spontaneously reported by participants. The burning sensation was assessed using a 100 mm Visual Analog Scale (VAS), where 0 indicated no sensation and 100 indicated the most intense burning or pain. Oral mucosal alterations were evaluated at seven sites, upper and lower lips, right and left buccal mucosa, hard palate, and dorsal and ventral tongue, using a 0–3 scoring system based on the Cosmetic, Toiletry, and Fragrance Association (CTFA) criteria [32]. The scoring was defined as follows: 0 indicated no observable changes; 1 indicated discoloration of the oral mucosa with mild edema; 2 indicated mucosal roughness with the beginning of lesions; and 3 indicated the presence of edema, ulcers, and sloughing of the oral mucosa. All adverse events were systematically monitored and documented according to a formal protocol, and each event was classified by severity as mild, moderate, or severe.

The assessment of soft tissue alterations was conducted by a dentist with over five years of experience, who had undergone both intra-examiner and inter-examiner calibration with an oral medicine specialist. Reliability was evaluated using Cohen’s Kappa coefficient (κ), which exceeded 0.80, and percent agreement values also surpassed 80%, supporting the consistency of the assessments.

User Satisfaction

The level of satisfaction was assessed on Day 1 (immediately after rinsing) and Day 14 using a questionnaire based on a 5-point Likert scale (1 = very dissatisfied, 2 = dissatisfied, 3 = neutral, 4 = satisfied, 5 = very satisfied). The satisfaction questionnaire was specifically designed for this study and underwent assessment for face and content validity through expert review. Five experts—including a prosthodontist, a general dentist, an oral medicine specialist, a pharmacist, and a microbiologist—independently evaluated the questionnaire to ensure the clarity, relevance, and comprehensiveness of the items. A trained researcher provided standardized instructions and clarification prior to the assessment, after which participants independently recorded their responses. The questionnaire evaluated participants’ perceptions across eight aspects: color, scent, texture, viscosity, taste, aftertaste, refreshing sensation, and duration, as well as overall satisfaction. It was self-administered under supervision to ensure clarity and completeness of responses.

### 2.5. Clinical Anticandidal and Antibacterial Efficacy

The concentrated oral rinse technique using NSS was performed on all participants before the use of mouthwash, and the CFU of *Candida* species was determined at baseline. Participants with CFU levels below 400 CFU/mL [28] were considered to be within the normal range and were included in the study.

Specimen Collection

Participants were instructed to refrain from eating and drinking for two hours prior to the appointment. The oral rinse sample was collected without dentures in the mouth, using 10 mL of sterile NSS, which the participant swished vigorously for 1 min before returning it to a sterile, sealed container.

Subsequently, the tongue coating sample was obtained using a sterile swab with a 5 mm diameter. The dorsal surface of the tongue was gently scraped from the circumvallate papilla to the tip of the tongue, covering a distance of at least 2 cm with a total of 15 strokes. The swab was then rotated 180 degrees, and the tongue was scraped an additional 15 times using the same method [33]. Finally, the swab was immediately agitated in the oral rinse sample for 1 min, and the container was then securely sealed and transferred to the biosafety cabinet in a 4 °C ice container.

Measurement of The Number of Oral Candida and Oral Bacteria

The oral rinse sampling technique [34] was conducted and slightly modified to determine oral bacteria and *Candida* in oral rinse samples. The oral rinse samples were then collected and transferred into 1 mL Eppendorf tubes, with 10 tubes used per sample, and centrifuged at 4000 rpm for 4 min using a CF-10 microcentrifuge (Daihan Scientific, Seoul, Republic of Korea). The process was performed on 4 °C in a biosafety cabinet. The supernatant was discarded, and the entire sediment was pooled into a single tube. The sample was then diluted 10-fold with NSS to achieve a final concentration of 1:1000 and cultured using the spread plate technique. Sabouraud Dextrose Agar (OxoidTM, Oxoid, Hampshire, UK) plates supplemented with 50 mg/L of chloramphenicol (SDCA) plates for oral *Candida* detection [35,36]. Columbia blood agar plates supplemented with 8 µg/mL of nystatin for inhibiting *Candida* and allow the growth of oral bacteria [37]. All cultures were done in triplicate and aerobically incubated at 37 °C for 24 h. All colonies grown on both agar plates were identified using Matrix Assisted Laser Desorption/Ionization—Time Of Flight Mass Spectrometry (MALDI-TOF MS) with the VITEK MS system (Biomerieux, Marcy l’Etoile, France). The CFU/mL for both oral *Candida* and bacterial amounts were then calculated.

### 2.6. Statistical Analysis

Statistical analyses were performed using GraphPad Prism software (version 9.0.0 Dotmatics Corp., Boston, MA, USA) and SPSS software (version 20.0.-IBM Corp., Armonk, NY, USA). Prior to analysis, data normality was assessed using the Shapiro–Wilk test.

Demographic data, satisfaction outcomes after 14 days, and clinical safety assessments were analyzed descriptively. One-way ANOVA was used to compare overall satisfaction, as well as satisfaction with color, scent, taste, viscosity, freshness, aftertaste, rinsing duration, and adverse reactions among the three mouthwash groups. Clinical anticandidal and antibacterial efficacies were analyzed using Tukey’s HSD post hoc test for pairwise comparisons between groups. A *p*-value < 0.05 was considered statistically significant.

## 3. Results

### 3.1. Demographic Characteristics of Participants

Of the 64 patients screened, 56 met the eligibility criteria and were enrolled in the study. Over the two-week study period, five participants were lost to follow-up: one from the NSS group, two from the CHX group, and one from the FCoMW group who failed to attend the scheduled follow-up visit. Additionally, one participant from the CHX group voluntarily withdrew before the compliance assessment. Consequently, a total of 51 participants completed the study and were included in the final analysis—comprising 17 in the NSS group, 16 in the CHX group, and 18 in the FCoMW group, as illustrated in Figure 1.

At the onset of the trial, the demographic characteristics of the 51 participants are presented in Table 2. The study population comprised predominantly males (68.63%), with females accounting for 31.37%. Most participants (58.82%) were aged 60–69 years, followed by 39.22% aged 70–79 and 1.96% aged 80 or older. Over half (56.86%) reported pre-existing medical conditions, the most commonly reported conditions being hypertension (41.18%), diabetes (29.41%), and hyperlipidemia (23.53%). Additionally, 98.04% of participants had no history of head and neck radiotherapy. None of the participants reported a history of oral hypersensitivity to mouthwash or its ingredients, and no cases of xerostomia were reported. Regarding oral hygiene habits, 72.55% brushed twice daily, and 60.78% had never used mouthwash. Among the 39.22% of participants with prior mouthwash use, 23.53% used it daily, and 35.29% preferred commercial products. Removable dentures were worn by 66.67% of participants. Among denture wearers (n = 34), most cleaned their dentures after every meal (51.43%) and removed them overnight (64.71%). The most frequently used cleaning methods were brushing with toothpaste (25.49%) and rinsing with water (25.49%). Overall, denture hygiene was classified as good in approximately half of the denture wearers (50.98%). Baseline demographic characteristics were comparable across the three intervention groups (all *p* > 0.05), except for underlying disease, which showed a statistically significant imbalance (*p* = 0.006).

### 3.2. Evaluation of Oral Health Parameters, Safety, and Satisfaction

Clinical Oral Health Parameters

At baseline, the plaque index was comparable across the three groups. However, a statistically significant difference in oral moisture levels was observed among the groups (*p* < 0.05), as presented in Table 3. After 14 days, all groups showed modest improvements in oral health parameters. The greatest mean reduction in plaque index was observed in the CHX group (−0.16 ± 0.13, 95% CI: −0.09–0.23, *p* < 0.001), followed by FCoMW (−0.09 ± 0.25, 95% CI: −0.046–0.09, *p* = 0.269) and NSS (−0.02 ± 0.15, 95% CI: −0.04–0.22, *p* = 0.677); however, these differences were not statistically significant (*p* = 0.092). Oral moisture levels increased in all groups, with FCoMW showing the highest mean increase (1.12 ± 3.21, 95% CI: −2.77–0.53, *p* = 0.169), followed by CHX (0.69 ± 2.27, 95% CI: −4.59–0.72, *p* = 0.246) and NSS (0.63 ± 2.09, 95% CI: −1.67–0.41, *p* < 0.217), with no significant differences among three groups (*p* = 0.833). The reliability of plaque index assessment was high, with intra-examiner κ = 0.949 (98% agreement) and inter-examiner κ = 0.854 (94% agreement).

Oral Safety Evaluation
Burning Sensation

A burning sensation was reported by 31.25% of participants in the CHX group (5 out of 16) on Day 1, as measured using the Visual Analog Scale (VAS), with a mean score of 0.95 ± 1.94. By Day 14, the same five participants continued to report this symptom, and the corresponding mean VAS score decreased to 0.31 ± 0.46. All reported burning sensations were of mild intensity, transient in nature, and resolved without the need for intervention. In contrast, no participants in the FCoMW or NSS groups reported experiencing a burning sensation at any time point. One-way ANOVA revealed significant differences in burning sensation among the groups on Day 1 (*p* = 0.014) and Day 14 (*p* = 0.02), as shown in Figure 2.
Oral Mucosal Alterations

Oral mucosal alterations were evaluated at seven anatomical sites using a 0–3 grading scale. No alterations were observed at baseline (Day 0), Day 1, or Day 14. All participants in all groups consistently scored 0 at all sites, indicating no mucosal irritation or damage. The reliability of oral mucosal assessment was high, with intra-examiner κ = 0.973 (98% agreement) and inter-examiner κ = 0.864 (90% agreement).

User Satisfaction

Participants’ satisfaction across eight aspects was assessed using a questionnaire administered on Day 1 (immediately following the intervention) and again on Day 14. Responses were recorded using a Likert scale to evaluate their perceptions over time, as shown in Figure 3 and Table 4. The FCoMW was generally well accepted, with high satisfaction ratings for color, scent, and freshness. On Day 1, the FCoMW group showed significantly lower mean color satisfaction compared with NSS and CHX (*p* < 0.05). However, by Day 14, mean scores slighly increased in the FCoMW group, and no significant differences were observed among the groups. Regarding scent, all mouthwash groups received high satisfaction ratings on both Day 1 and Day 14. No statistically significant differences in mean scent scores were observed among the groups (*p* > 0.05). Freshness ratings remained consistently high across both time points, particularly in the FCoMW group. No significant differences were observed among groups in terms of freshness satisfaction on Day 1 and Day 14. By Day 14, overall satisfaction reached 88.89%, the highest among the three groups. FCoMW consistently received positive feedback, with no reported dissatisfaction in several categories, particularly in scent, taste and freshness. In terms of taste satisfaction scores were highest for NSS, moderate for FCoMW, and lowest for CHX. Although the taste scores of FCoMW were slightly lower tha those of NSS, they were higher than those of CHX. However, no significant differences were observed among the groups. In contrast, a few participants reported dissatisfaction with viscosity. Satisfaction with viscosity was significantly lower in the FCoMW group compared to CHX and NSS at both time points (*p* = 0.002 on Day 1; *p* < 0.001 on Day 14). Regarding rinsing duration, some participants in the FCoMW and NSS groups reported dissatisfaction on Day 1. By Day 14, only one participant in the NSS group remained dissatisfied. Although the CHX group showed significantly higher satisfaction with rinsing duration on Day 1 (*p* = 0.008), this difference was not statistically significant by Day 14 (*p* = 0.160). With respect to aftertaste, 25% of participants in the CHX group consistently rated their aftertaste satisfaction as low (scores 1–2) on both Day 1 and Day 14. In contrast, the FCoMW group demonstrated more favorable outcomes, with only 5.56% reporting low satisfaction on Day 1 and none by Day 14. Consequently, the CHX group exhibited the lowest aftertaste satisfaction among the three mouthwash formulations.

### 3.3. Clinical Anticandidal and Antibacterial Efficacy

The Antibacterial Efficacy

The results demonstrated a statistically significant reduction (*p* < 0.0001) in oral bacteria levels after 14 days of application of 0.12% *w*/*v* CHX (CHX D14) and FCoMW (FCoMW D14) compared to the baseline levels (CHX D0, FCoMW D0). Specially, a statistically significant reduction (*p* < 0.0001) in oral bacteria levels after 14 days of application of 0.12% *w*/*v* CHX (CHX D14; median = 0, interquartile range (IQR) = 0–1.348) and FCoMW (FCoMW D14; median = 2.226, IQR = 2.191–2.276) compared to the baseline levels (CHX D0; median = 3.604, IQR = 3.352–4.110, FCoMW D0; median = 3.708, IQR = 3.290–3.899). Additionally, significant differences (*p* < 0.0001) were observed when comparing bacterial levels after 14 days of 0.12% *w*/*v* CHX use (CHX D14; median = 0, IQR = 0–1.348) to those of FCoMW (FCoMW D14; median = 2.226, IQR = 2.191–2.276) and NSS (NSS D14; median = 3.789, IQR = 3.573–3.866), as well as between FCoMW (FCoMW D14; median = 2.226, IQR = 2.191–2.276) and NSS (NSS D14; median = 3.789, IQR = 3.573–3.866). Median bacterial level of the NSS baseline group (NSS D0) was 3.660, IQR = 3.396–4.039 as illustrated in Figure 4.

The Anticandidal Efficacy

The results demonstrated a statistically significant reduction (*p* < 0.0001) in oral *Candida* levels from baseline (CHX D0; median = 1.459, IQR = 1.080–1.701) to Day 14 (CHX D14; median = 0.093, IQR = 0–0.767) following the use of 0.12% *w*/*v* CHX mouthwash. A similar significant reduction (*p* < 0.0001) was observed in the FCoMW group between baseline (FCoMW D0; median value = 1.533, IQR = 0.979–1.780) and Day 14 (FCoMW D14; median = 0.580, IQR = 0.217–0.863). Additionally, on Day 14, both CHX (CHX D14) and FCoMW (FCoMW D14) showed significantly lower *Candida* levels (*p* < 0.0001) compared to NSS group (NSS D14; median = 1.374, IQR = 1.066–1.595). However, no statistically significant difference was found between FCoMW (FCoMW D14; median = 0.580, IQR = 0.217–0.863) and CHX (CHX D14; median = 0.093, IQR = 0–0.767) on Day 14, as shown in Figure 5. Median *Candida* level of NSS baseline group (NSS D0) was 1.451, IQR = 1.110–1.620.

## 4. Discussion

The null hypothesis was partially rejected, as FCoMW demonstrated oral safety comparable to NSS while providing superior user acceptability and anticandidal efficacy equivalent to CHX, with no significant differences observed in mucosal safety outcomes.

These findings suggest that FCoMW may represent a safe, effective, and well-accepted alternative to conventional chemical mouthwashes for older adults.

Traditional oil pulling with VCO is often hindered by several limitations, including its high viscosity, greasy texture, and the prolonged rinsing time required to achieve therapeutic efficacy [16]. The development of the FCoMW represents a significant advancement in addressing these drawbacks. The development of FCoMW constitutes a significant advancement in overcoming these limitations. By integrating formulation refinement with demonstrated clinical performance in an older population, this study highlights the potential of FCoMW as a practical and acceptable alternative to traditional oil pulling for maintaining oral hygiene in older individuals.

### 4.1. Antibacterial and Antifungal Efficacies

Clinical assessment through a randomized controlled trial demonstrated that FCoMW significantly reduced oral *Candida* loads after 14 days of use. Its antimicrobial efficacy was statistically comparable to that of CHX, reinforcing prior in vitro evidence of VCO’s capacity to inhibit microbial adhesion and disrupt biofilm formation [26]. Notably, no prior clinical studies have directly compared the anticandidal efficacy of coconut oil with that of CHX, underscoring the novelty and clinical relevance of the present investigation.

Additionally, a statistically significant difference in bacterial levels was observed between the FCoMW and NSS groups after 14 days. This finding aligns with the results of a meta-analysis, which demonstrated that oil pulling with coconut oil may have a beneficial effect in reducing salivary bacterial colony counts when compared to distilled water [38]. The antimicrobial properties of VCO are primarily attributed to lauric acid and monolaurin. Both compounds possess amphiphilic characteristics, enabling interaction with microbial membranes. The lipophilic lauryl chains facilitate Van der Waals interactions with non-polar regions of microbial cell walls, while the hydroxyl and carbonyl groups enhance hydrophilicity. This duality allows for membrane insertion, increased permeability, and cytoplasmic leakage, ultimately leading to microbial cell death. In Gram-negative bacteria, this mechanism disrupts the lipid bilayer, whereas in Gram-positive bacteria, it compromises cell wall integrity. Additionally, monolaurin impairs quorum sensing and biofilm formation. In *C. albicans*, it interferes with morphological transitions and downregulates key virulence-associated genes such as *ALS3* and *HWP1*, thereby attenuating fungal pathogenicity [39]. Moreover, several hypotheses have been proposed. The mechanical action of oil-pulling facilitates emulsification between oil and saliva, producing a viscous emulsion capable of entrapping oral microorganisms. This emulsified mixture promotes the encapsulation of bacteria-laden epithelial cells within lipid micelles, thereby aiding their removal from the oral cavity. In addition, oil-pulling stimulates salivary flow, which supports the natural cleansing and antimicrobial functions of saliva. The process also leads to the formation of a lipid-enriched pellicle on tooth surfaces, increasing surface hydrophobicity and reducing bacterial adhesion and colonization [40]. Supporting this mechanism, nano- and micro-sized lipid droplets have been observed adhering to tooth surfaces after a 30-s oil rinse and remaining in place for several hours [41]. Reinforcing the clinical relevance of these findings, participants in the present study were instructed to retain the FCoMW formulation intraorally for five min. This protocol was adapted from Chanpa et al. [26], who confirmed that CoMW effectively inhibited *C. albicans* after comparable exposure, as verified by quantitative colony counts.

### 4.2. Safety and Biocompatibility

FCoMW demonstrated excellent biocompatibility with oral tissues. Mucosal examinations conducted at seven anatomical sites revealed no observable adverse tissue reactions among participants.

Furthermore, none of the FCoMW participants reported burning sensations, whereas a proportion of CHX users experienced discomfort. This finding is consistent with previously documented adverse effects associated with CHX, including mucosal irritation, altered taste sensation (dysgeusia), and disruption of the oral microbiome [6,42]. Moreover, this observation is also in agreement with previous study on oil pulling with VCO, in which users reported no signs of mucosal irritation following two weeks of continuous use [16]. The absence of such effects in the FCoMW group emphasizes its favorable safety profile for sustained use, particularly among individuals requiring ongoing oral antimicrobial support, such as denture wearers who are at elevated risk for *Candida*-associated denture stomatitis.

Although no adverse events were reported in this study, the long-term safety of oil-based rinses warrants caution. Edible oils may contain environmental contaminants such as polycyclic aromatic hydrocarbons (PAHs), plasticizers, pesticides, solvents, and heavy metals. While systemic absorption through oral rinsing is presumed minimal, these risks may be clinically significant for medically vulnerable populations. Therefore, ensuring oil purity and adherence to stringent manufacturing standards is essential for product safety. Furthermore, although uncommon, serious complications such as exogenous lipoid pneumonia have been documented in association with chronic aspiration or inhalation of oil-based substances. For instance, Kuroyama et al. [43] reported two cases of lipoid pneumonia in individuals who engaged in routine oil pulling with sesame oil for several months, both presenting with radiographic ground-glass opacities and lipid-laden alveolar macrophages. Children are particularly susceptible to aspiration-related complications from oil-based rinses due to immature swallowing mechanisms and limited ability to expectorate. This concern further justifies the conduct of the present trial in older adults, who can be more reliably instructed and monitored during mouthwash use, thereby minimizing aspiration-related risks. Consequently, the use of oil-based mouthwashes sholud be avoided in patients with dysphagia, neurologic disorders, or other conditions in which the risk of aspiration cannot be reliably excluded.

### 4.3. Effects on Oral Moisture and Plaque Control

VCO has been proposed as a natural remedy for xerostomia due to its emollient properties and its ability to form a protective lipid film over mucosal surfaces, thereby reducing moisture loss. Recent study suggest that coconut oil-based rinses may enhance oral moisture and improve subjective comfort, particularly in individuals with salivary gland dysfunction or those undergoing post-radiation therapy [13]. In the present study, the FCoMW group demonstrated the greatest improvement in oral moisture levels, although the difference did not reach statistical significance when compared between groups. In terms of plaque reduction, only the 0.12% CHX group showed a statistically significant within-group decrease in plaque index after 14 days of use, with no significant differences observed between groups. This outcome contrasts with previous reports highlighting the efficacy of VCO in reducing plaque accumulation when assessed over a 30-day period [44]. A plausible explanation for this discrepancy is the relatively low baseline plaque index observed across all groups, which likely reflects the participants’ generally good oral hygiene behaviors, including regular toothbrushing (at least twice daily), consistent denture care, and routine dental visits every six months. Such favorable baseline conditions may have limited the potential to detect clinically meaningful improvements. Furthermore, the 14-day intervention period may have been insufficient to reveal measurable differences in plaque accumulation, as evaluations of plaque index are generally recommended over a minimum of 30 days for a more reliable assessment [15,44,45].

### 4.4. User Satisfaction and Acceptability

User satisfaction plays a crucial role in promoting adherence to oral hygiene practices. In the present study, participants reported a high level of satisfaction with the taste, aroma, and freshness of FCoMW. Although satisfaction scores related to viscosity and rinsing duration were slightly lower than those for CHX and NSS, these factors did not appear to negatively affect overall compliance. Notably, the overall satisfaction score for the FCoMW group remained high on Day 14 at 88.89%, suggesting that viscosity and rinsing duration were not major barriers to user acceptance. These findings highlight the importance of optimizing both flavor and emulsion texture to ensure long-term adherence to oil-based oral care regimens [26]. Aftertaste is an important determinant of patient adherence to mouthwash use, particularly in the context of long-term preventive oral care. In the present study, CHX consistently yielded lower aftertaste satisfaction scores, with 25% of participants reporting unpleasant aftertaste (scores 1–2) on both Day 1 and Day 14. This observation is consistent with previous literature indicating that CHX often produces a lingering bitter or metallic aftertaste and may transiently impair taste perception [7]. By contrast, FCoMW demonstrated more favorable sensory outcomes, suggesting that it may provide a more acceptable alternative and potentially improve user adherence in long-term preventive oral care.

### 4.5. Clinical Implications

The outcomes of this study support the use of FCoMW as a safe, effective, and acceptable oil-based alternative to CHX mouthwash for older adults. Its broad-spectrum antimicrobial activity, excellent mucosal compatibility, and favorable user acceptance highlight its potential for long-term preventive oral care, particularly in settings where tolerability is critical. The incorporation of essential oils enhanced the sensory attributes of the formulation, an important factor for promoting sustained adherence in real-world settings.

Clinically, FCoMW may be especially relevant for elderly denture wearers at risk of Candida-associated stomatitis, patients with mucosal sensitivity who cannot tolerate chemical antiseptics, individuals with xerostomia requiring emollient support, and immunocompromised populations who require a safe preventive or antifungal mouthwash that avoids the adverse effects associated with CHX.

### 4.6. Limitations and Future Perspectives

This study has several limitations. First, the relatively small sample size and single-center design, restricted to older adults attending a university dental clinic, may limit the generalizability of the findings. In addition, participant blinding was not feasible due to the distinct characteristics of the tested mouthwashes, although evaluator blinding was maintained to minimize assessment bias.

Future research should confirm these findings in larger, multi-center studies that include more diverse populations, including individuals with systemic immunocompromise or xerostomia. Additionally, long-term clinical studies are warranted to assess the durability of microbial suppression, potential shifts in the oral microbiota, and the impact of FCoMW on periodontal health parameters such as gingival inflammation and plaque composition. In addition, with the increasing prevalence of non-*albicans Candida* (NAC) species-accounting for nearly one-third of isolates and exhibiting high azole resistance, particularly *Candida krusei* [46]-further studies are warranted to investigate the antifungal efficacy of FCoMW against these strains.

## 5. Conclusions

In a 14-day randomized controlled trial, FCoMW demonstrated anticandidal efficacy comparable to 0.12% *w*/*v* CHX, with no adverse effects and high user satisfaction. These findings support FCoMW as a safe, well-tolerated, and efficacious natural mouthwash with anticandidal activity, offering a promising long-term alternative for preventive dental care, particularly for elderly individuals, denture wearers, and patients with mucosal sensitivity or those at risk of oral candidiasis, for whom prolonged CHX use is limited by adverse effects.

## Figures and Tables

**Figure 1 healthcare-13-02941-f001:**
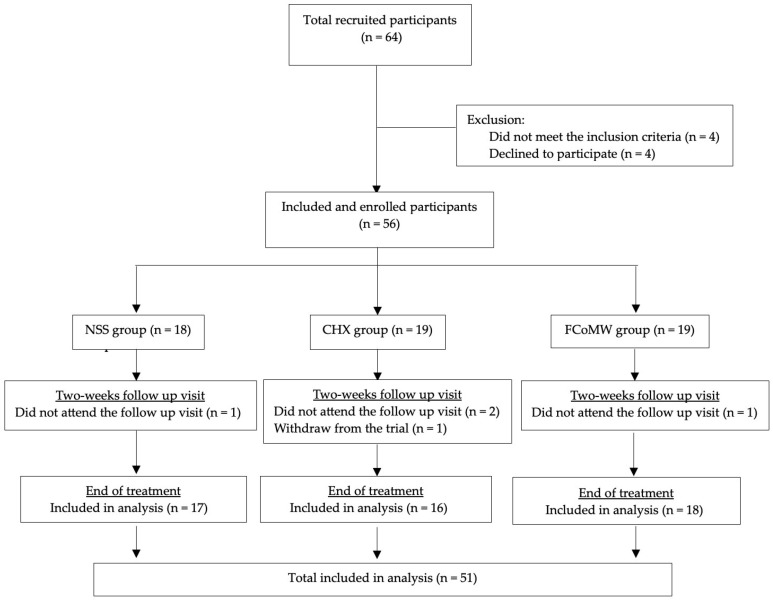
CONSORT flow diagram of participant recruitment, allocation, follow-up, and analysis. Abbreviations: NSS, normal saline solution; CHX, chlorhexidine; FCoMW, flavor-enhanced coconut oil-based mouthwash.

**Figure 2 healthcare-13-02941-f002:**
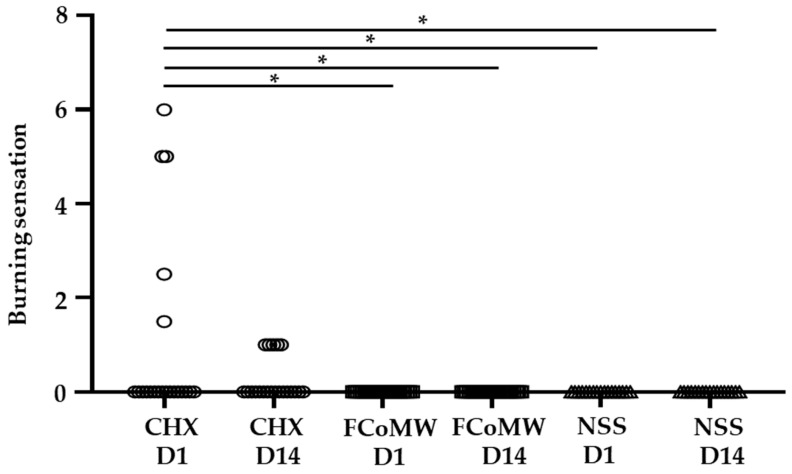
Comparison of the burning sensation scores among participants using FCoMW, CHX, and NSS. Data are illustrated across six groups including at the start of the trial (Day 1; CHX D1, FCoMW D1, and NSS D1) and the end of the trial (Day 14; CHX D14, FCoMW D14, and NSS D14). * denotes statistically significant difference (*p* < 0.01) as determined by Tukey’s HSD test. Abbreviations: CHX, chlorhexidine; FCoMW, flavor-enhanced coconut oil-based mouthwash; NSS, normal saline solution; D1, Day 1; D14, Day 14.

**Figure 3 healthcare-13-02941-f003:**
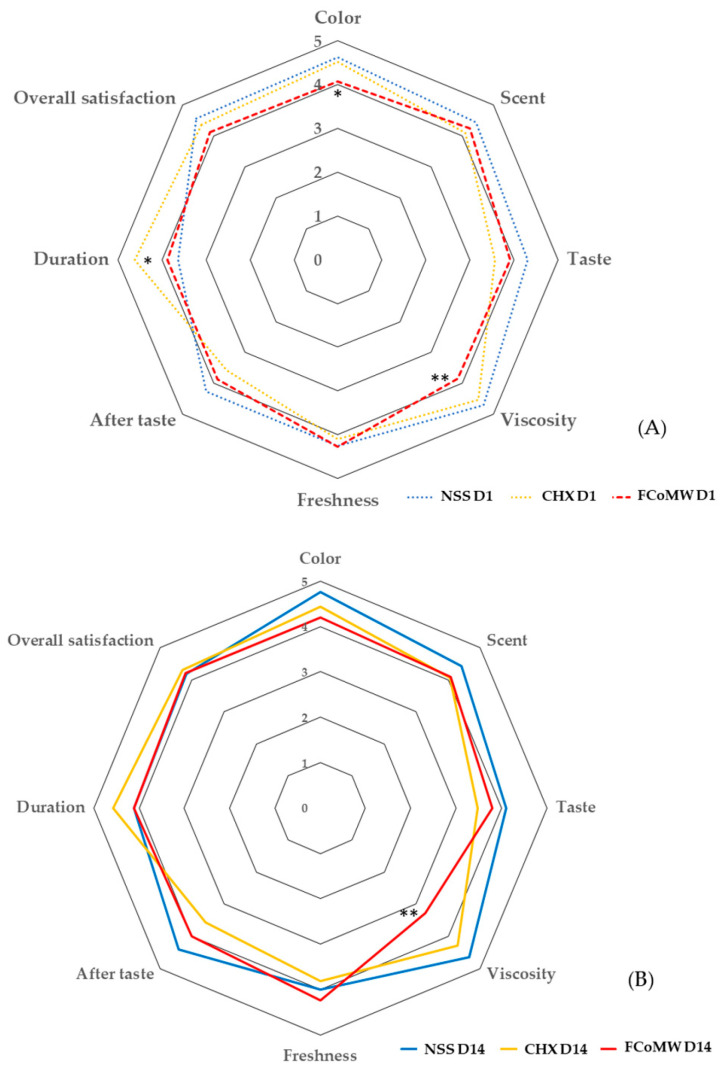
Mean satisfaction scores for eight aspects of three different mouthwash groups on (**A**) Day 1 and (**B**) Day 14. Denote statistically significant differences were determined using one-way ANOVA: *p* < 0.05 (*), and *p* < 0.001 (**). Scores indicate satisfaction levels: 5 = very satisfied, 4 = satisfied, 3 = neutral, 2 = unsatisfied, and 1 = very unsatisfied. Abbreviations: NSS, normal saline solution; CHX, chlorhexidine; FCoMW, flavor-enhanced coconut oil-based mouthwash.

**Figure 4 healthcare-13-02941-f004:**
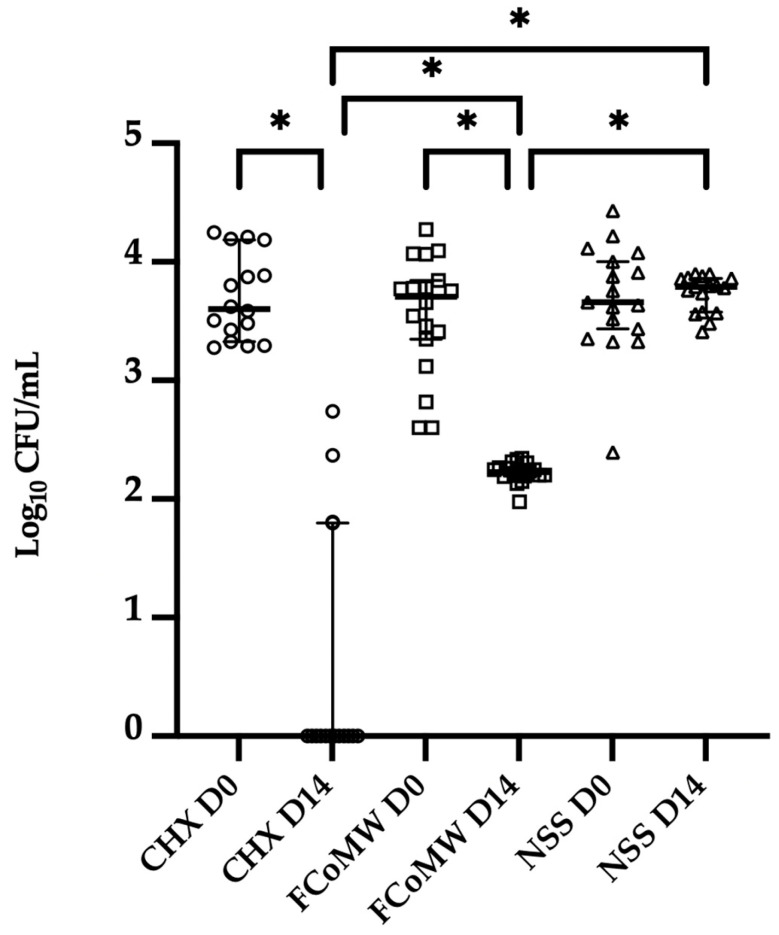
Oral bacterial quantification on Day 0 and Day 14 following the application of three mouthwashes: FCoMW, 0.12% *w*/*v* CHX (positive control), and NSS (negative control). Horizontal bars represent the median value of each treatment group. * denotes the significant difference at *p* < 0.0001 as determined by Tukey’s HSD test. Abbreviations: CHX, chlorhexidine; FCoMW, flavor-enhanced coconut oil-based mouthwash; NSS, normal saline solution; D0, Day 0; D14, Day 14.

**Figure 5 healthcare-13-02941-f005:**
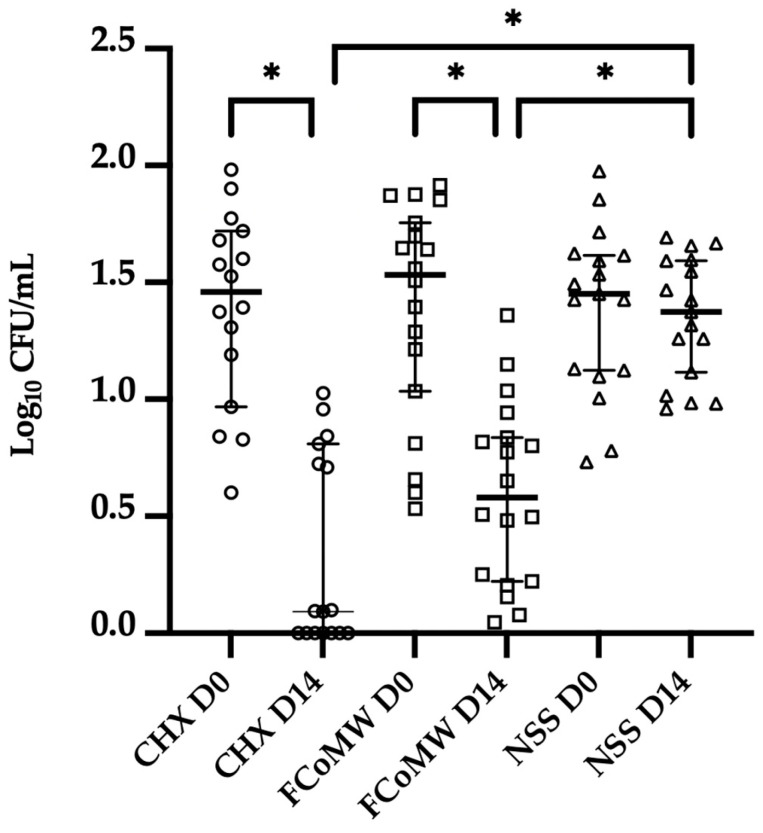
Oral *Candida* quantification on Day 0 and Day 14 following the application of three mouthwashes: FCoMW, 0.12% *w*/*v* CHX (positive control) and NSS (negative control). Horizontal bars represent the median value of each treatment group. * denotes a statistically significant difference at *p* < 0.0001 as determined by Tukey’s HSD test. Abbreviations: CHX, chlorhexidine; FCoMW, flavor-enhanced coconut oil-based mouthwash; NSS, normal saline solution; D0, Day 0; D14, Day 14.

**Table 1 healthcare-13-02941-t001:** Eligibility criteria for study participants.

Inclusion Criteria	Exclusion Criteria
Aged over 60 years In good general health with well-controlled medical conditions (ASA physical status I–II) Independent older adults Normal swallowing function *Candida* species detected by culture from oral rinse samples, with colony counts not exceeding 400 CFU	Presence of significant Medical or oral abnormalities that could interfere with the study Intolerance or hypersensitivity to study interventions Current smoking or smoking cessation within the past year Use of topical or systemic antifungal medications within one month prior to enrollment

ASA = American Society of Anesthesiologists; CFU = colony-forming unit.

**Table 2 healthcare-13-02941-t002:** Demographic and oral health characteristics of study participants.

Variables		NSS % (n = 17)	0.12% *w*/*v* CHX % (n = 16)	FCoMW % (n = 18)	Total % (n = 51)	*p*-Value
1. Gender	Female	29.40 (5)	25.00 (4)	38.88 (7)	31.37 (16)	0.669
	Male	70.60 (12)	75.00 (12)	61.11 (11)	68.63 (35)	
2. Age	60–69	58.80 (10)	68.75 (11)	50.00 (9)	58.82 (30)	0.431
	70–79	35.30 (6)	31.25 (5)	50.00 (9)	39.22 (20)	
	>80	5.90 (1)	0.00 (0)	0.00 (0)	1.96 (1)	
3. Underlying disease	Had	64.70 (11)	31.25 (5)	72.22 (13)	56.86 (29)	0.006 *
	Hypertension	47.10 (8)	31.25 (5)	44.44 (8)	41.18 (21)	
	Diabetes	5.90 (1)	25.00 (4)	27.78 (5)	29.41 (15)	
	Hyperlipidemia	11.80 (2)	18.75 (3)	38.89 (7)	23.53 (12)	
	Others	29.40 (5)	12.50 (2)	11.11 (2)	17.65 (9)	
	None	35.30 (6)	68.75 (11)	27.78 (5)	43.14 (22)	
4. History of head and neck radiotherapy	No	100.00 (17)	100.00 (16)	94.44 (17)	98.04 (50)	0.393
	Yes	0.00 (0)	0.00 (0)	5.56 (1)	1.96 (1)	
5. History of mouthwash allergy	No	100.00 (17)	100.00 (16)	100.00 (18)	100.00 (51)	NA
	Yes	0.00 (0)	0.00 (0)	0.00 (0)	0.00 (0)	
6. Xerostomia	No	100.00 (17)	100.00 (16)	100.00 (18)	100.00 (51)	NA
	Yes	0.00 (0)	0.00 (0)	0.00 (0)	0.00 (0)	
7. Frequency of brushing teeth	<2 times/day	0.00 (0)	18.75 (3)	5.56 (1)	7.84 (4)	0.691
	2 times/days	82.40 (14)	62.50 (10)	72.22 (13)	72.55 (37)	
	>2 times/days	17.60 (3)	18.75 (3)	22.22 (4)	19.61 (10)	
8. Using mouthwash	No	58.80 (10)	56.25 (9)	66.67 (12)	60.78 (31)	0.804
	Yes	41.20 (7)	43.75 (7)	33.33 (6)	39.22 (20)	
8.1. Frequency	Everyday	17.64 (3)	25.00 (4)	11.11 (2)	23.53 (12)	
	Sometimes	23.53 (4)	18.75 (3)	22.22 (4)	15.69 (8)	
8.2. Type of mouthwash	Commercial mouthwash	35.29 (6)	43.75 (7)	27.78 (5)	35.29 (18)	
	Therapeutic mouthwash	5.88 (1)	0.00 (0)	5.56 (1)	3.92 (2)	
9. Presence of removable denture	No	41.18 (7)	25.00 (4)	33.33 (6)	33.33 (17)	0.230
	Yes	58.82 (10)	75.00 (12)	66.67 (12)	66.67 (34)	
9.1 Denture’s age (n = 34)	<5 years	23.53 (4)	50.00 (8)	33.33 (6)	35.29 (18)	
	≥5 years	35.29 (6)	25.00 (4)	33.33 (6)	31.37 (16)	
9.2 Frequency of denture cleaning (n = 34)	No	0.00 (0)	0.00 (0)	0.00 (0)	0.00 (0)	
	Once/day	0.00 (0)	12.50 (2)	5.56 (1)	5.88 (3)	
	Twice/day	35.29 (6)	25.00 (4)	16.67 (3)	25.49 (13)	
	After every meal	23.53 (4)	37.50 (6)	44.44 (8)	51.43 (18)	
9.3 Method of cleaning (n = 34)	Toothbrush + toothpaste	35.29 (6)	18.75 (3)	22.22 (4)	25.49 (13)	
	Toothbrush + soap	5.88 (1)	18.75 (3)	22.22 (4)	15.69 (8)	
	Water	17.65 (3)	37.50 (6)	22.22 (4)	25.49 (13)	
9.4 Overnight removal (n = 34)	No	0.00 (0)	0.00 (0)	5.56 (1)	1.96 (1)	
	Yes	58.82 (10)	75.00 (12)	61.11 (11)	64.71 (33)	
9.5 Denture hygiene (n = 34)	Good	41.18 (7)	68.75 (11)	44.44 (8)	50.98 (26)	
	Poor	17.65 (3)	6.25 (1)	22.22 (4)	15.69 (8)	

Other underlying medical conditions included hypothyroidism and cardiovascular disease. Differences in demographic and oral health characteristics across the three groups were assessed using the Chi-square test or Fisher’s exact test, as appropriate. A *p*-value < 0.05 was considered statistically significant. Statistical significance is indicated as follows: * *p* < 0.01. Abbreviations: NSS, normal saline solution; CHX, chlorhexidine; FCoMW, flavor-enhanced coconut oil-based mouthwash.

**Table 3 healthcare-13-02941-t003:** Comparison of plaque index, and oral moisture levels (mean ± SD) among three mouthwash groups at baseline and Day 14, including mean changes.

Parameter	NSS (n = 17)	0.12% *w*/*v* CHX (n = 16)	FCoMW (n = 18)	*p*-Value (Between Group)
**Plaque index**				
Baseline	1.22 ± 0.22	1.38 ± 0.21	1.22 ± 0.31	0.098
Day 14	1.20 ± 0.24	1.23 ± 0.26	1.13 ± 0.31	0.578
Mean changes (Δ) (Baseline-Day 14)	−0.02 ± 0.15	−0.16 ± 0.13	−0.09 ± 0.25	0.092
*p*-value (within group)	0.677	<0.001 **	0.269	
**Oral moisture level**				
Baseline	30.09 ± 1.99	30.12 ± 1.38	28.40 ± 2.93	0.042 *
Day 14	30.72 ± 0.79	30.81 ± 1.65	29.51 ± 2.11	0.040 *
Mean changes (Δ) (Baseline-Day 14)	0.63 ± 2.09	0.69 ± 2.27	1.12 ± 3.21	0.833
*p*-value (within group)	0.217	0.246	0.169	

*p*-values between groups were determined by one-way ANOVA; within-group comparisons were analyzed with paired *t*-tests. Statistical significance is indicated as follows: * *p* < 0.05, ** *p* < 0.001. Abbreviations: NSS, normal saline solution; CHX, chlorhexidine; FCoMW, flavor-enhanced coconut oil-based mouthwash; SD, standard deviation.

**Table 4 healthcare-13-02941-t004:** Satisfaction levels for three different mouthwash groups on Day 1 and Day 14.

Parameters	Day	Levels of Satisfaction	NSS (n = 17), %(n)		0.12% *w*/*v* CHX (n = 16), %(n)		FCoMW (n = 18), %(n)	
			Mean ± S.D.	Median (IQR)	Mean ± S.D.	Median (IQR)	Mean ± S.D.	Median (IQR)
Color	1		4.65 ± 0.61 ^a^	5 (4–5)	4.69 ± 0.48 ^b^	5 (4–5)	4.17 ± 0.79 ^a,b^	4 (3.75–5)
		4–5	94.12% (16)		100 (16)		77.78 (14)	
		3	6.25% (1)		0 (0)		22.22 (4)	
		1–2	0 (0)		0 (0)		0 (0)	
	14		4.70 ± 0.59	5 (4–5)	4.50 ± 0.73	5 (4–5)	4.28 ± 0.84	5 (3–5)
		4–5	94.12 (16)		77.78 (14)		72.22 (13)	
		3	5.88 (1)		12.5 (2)		27.78 (5)	
		1–2	0 (0)		0 (0)		0 (0)	
Scent	1		4.35 ± 0.70	4 (4–5)	4.25 ± 1.00	5 (3.25–5)	4.22 ± 0.64	4 (4–5)
		4–5	88.24 (15)		75.00 (12)		88.89 (16)	
		3	11.76 (2)		18.75 (3)		11.11 (2)	
		1–2	0 (0)		6.25 (1)		0 (0)	
	14		4.35 ± 0.79	5 (4–5)	4.50 ± 0.73	5 (4–5)	4.11 ± 0.90	4 (3–5)
		4–5	82.35 (14)		88.24 (15)		66.67 (12)	
		3	17.65 (3)		12.50 (2)		33.33 (6)	
		1–2	0 (0)		0 (0)		0 (0)	
Viscosity	1		4.71 ± 0.69 ^c^	5 (5–5)	4.63 ± 0.73 ^d^	5 (4–5)	3.56 ± 0.98 ^c,d^	3 (3–4.25)
		4–5	88.24 (15)		93.75 (15)		44.45 (8)	
		3	11.76 (2)		6.25 (1)		44.44 (8)	
		1–2	0 (0)		0 (0)		11.11 (2)	
	14		4.35 ± 0.93 ^e^	5 (5–5)	4.44 ± 0.73 ^f^	5 (4–5)	3.22 ± 0.81 ^e,f^	3 (3–4)
		4–5	82.35 (14)		87.5 (14)		33.33 (6)	
		3	11.76 (2)		12.50 (2)		50.00 (9)	
		1–2	5.88 (1)		0 (0)		16.67 (3)	
Freshness	1		4.18 ± 0.81	4 (4–5)	4.19 ± 0.83	4 (3.25–5)	4.11 ± 0.83	4 (3–5)
		4–5	88.24 (15)		75.00 (12)		72.22 (13)	
		3	5.88 (1)		25.00 (4)		27.78 (5)	
		1–2	5.88 (1)		0 (0)		0 (0)	
	14		4.00 ± 0.87	4 (4–5)	4.00 ± 0.97	4 (3–5)	4.28 ± 0.75	4 (4–5)
		4–5	76.47 (13)		68.75 (11)		83.33 (15)	
		3	17.65 (3)		25.00 (4)		16.67 (3)	
		1–2	5.88 (1)		6.25 (1)		0 (0)	
Taste	1		4.47 ± 0.62	5 (4–5)	3.69 ± 1.35	4 (3–5)	3.94 ± 0.80	4 (3–5)
		4–5	94.12 (16)		68.75 (11)		66.67 (12)	
		3	5.88 (1)		11.11 (2)		33.33 (6)	
		1–2	0 (0)		18.75 (3)		0 (0)	
Taste	14		4.12 ± 0.78	5 (4–5)	3.63 ± 1.50	4 (2.25–5)	3.78 ± 0.80	4 (3–4.25)
		4–5	88.24 (15)		56.25 (9)		55.56 (10)	
		3	5.88 (1)		18.75 (3)		44.44 (8)	
		1–2	5.88 (1)		25.00 (4)		0 (0)	
Aftertaste	1		4.24 ± 0.75	4 (4–5)	3.75 ± 1.39	4 (2.25–5)	3.89 ± 0.83	4 (3–4.25)
		4–5	82.35 (14)		62.5 (10)		72.22 (13)	
		3	17.65 (3)		12.50 (2)		22.22 (4)	
		1–2	0 (0)		25.00 (4)		5.56 (1)	
	14		4.24 ± 0.75	4 (3–5)	3.63 ± 1.45	4 (2.25–5)	4.00 ± 0.84	4 (3–4.25)
		4–5	82.35 (14)		62.5 (10)		66.67 (12)	
		3	17.65 (3)		12.50 (2)		33.33 (6)	
		1–2	0 (0)		25.00 (4)		0 (0)	
Duration	1		3.71 ± 1.36 ^g^	4 (3–5)	4.75 ± 0.57 ^g,h^	5 (5–5)	3.72 ± 1.02 ^h^	4 (3–5)
		4–5	64.71 (11)		93.75 (15)		55.56 (10)	
		3	17.65 (3)		6.25 (1)		33.33 (6)	
		1–2	17.65 (3)		0 (0)		11.11 (2)	
	14		4.12 ± 1.11	4 (3.5–5)	4.63 ± 0.72	5 (4.25–5)	4.11 ± 0.90	4 (3–5)
		4–5	76.47 (13)		87.5 (14)		66.67 (12)	
		3	17.65 (3)		12.50 (2)		33.33 (6)	
		1–2	5.88 (1)		0 (0)		0 (0)	
Overall satisfaction	1		4.53 ± 0.62	5 (4–5)	4.31 ± 0.95	5 (4–5)	4.06 ± 0.80	4 (3–5)
		4–5	94.12 (16)		81.25 (13)		72.22 (13)	
		3	5.88 (1)		12.50 (2)		27.78 (5)	
		1–2	0 (0)		6.25 (1)		0 (0)	
	14		4.18 ± 0.73	4 (4–5)	4.38 ± 0.72	4.5 (4–5)	4.33 ± 0.69	4 (4–5)
		4–5	82.35 (14)		87.5 (14)		88.89 (16)	
		3	17.65 (3)		12.50 (2)		11.11 (2)	
		1–2	0 (0)		0 (0)		0 (0)	

Scores indicate satisfaction levels as follows: 4–5, Satisfied–very satisfied; 3, Neutral; 1–2, Unsatisfied-very unsatisfied. Different letters (a–h) represent significant differences according to Tukey’s HSD test. Abbreviations: NSS, normal saline solution; CHX, chlorhexidine; FCoMW, flavor-enhanced coconut oil-based mouthwash.

## Data Availability

The data are available on request from the corresponding author due to privacy restrictions.

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
