# Peer review of "A Randomized Trial in Older Adults of a Flavor-Enhanced Coconut Oil-Based Mouthwash: Clinical Safety, Antimicrobial Efficacy, and User Satisfaction"

_healthcare, 2025, doi:10.3390/healthcare13222941_

Round 1
Reviewer 1 Report
Comments and Suggestions for Authors
Review 1
A Randomized Trial in Older Adults of a Flavor-Enhanced Coconut Oil-Based Mouthwash: Clinical Safety, Antimicrobial Efficacy, and User Satisfaction
The authors presente a study aimed in the developing of a clinical safe coconut oil-based mouthwash (FCoMW), antimicrobial effective and satisfactory to its users- an older population. They use a negative control – saline solution and a positive control- CHX and compare their results after a 14 days use. Virgin coconut oil (VCO) has become popular for general health promotion and several studies show also that can improve oral health as na adjunctive oral hygiene care. Also This kind of approach is very much tested as CHX long-term use, is known to have several side effects.
The study is well designed and presented and I find that only minor changes should/could be made to enhance its scientific value.
Minor changes/points to adress
Abstract
Line 25- “……coconut oil-based mouthwash (FCoMW)…….. “should be flavor-enhanced coconut oil-based mouthwash (FCoMW).
Introduction
Line 48- Candida albicans— should be in italic
Line 49-“xerostomia” is not a condition is a sensation, I would use hiposalivation
Line 52- “Candida species”- should be in italic
Line 74- “gingivitis reduction [13-15], halitosis reduction” ( rephrase it)
Line 79- “…….oral malodor, its use in…….” should be “…..oral malodor. Nevertheless, its use in……”
Line 84- “demonstrating non-inferiority to 100,000 units/mL of nystatin (p>0.05) [23]”. Should rephrase, or use more scientific background- I do not understand the meaning.
Materials and Methods
Line 117 – “One % w/v…...” should be One percent
Line 154- “NSS group…” first time said- should be normal saline solution (NSS)
Line 190 – “Oral mucosal alterations was evaluated…” should be …..were evaluated
Line 208- For the NSS group, after the 21-day washout period, participants were randomly assigned to either the FCoMW mouthwash group or the CHX mouthwash group for further treatment.
This paragraph leaves space to ask question about missing results. I Understand it was done but, in that case, should be also presented in the figure 1
Line 234-240- to long sentence-break it
Line 234-240- to long sentence-break it
Results
Line 308- “…..FcoMW…” should be FCoMW
Discussion
- When discussing the results from Plaque index or bacterial index should be noted that the FCoMW group, if treated with nystatin would also show a bacterial growth block, i.e S. mutans that introduce a bias in the results.
You migth consider to introduce the Reference:
Nystatin-treated group to detect growth of oral nbacteria, but this also blocks s. mjutans growth…… Alomeir, N., Zeng, Y., Fadaak, A., Wu, T. T., Malmstrom, H., & Xiao, J. (2023). Effect of Nystatin on Candida albicans - Streptococcus mutans duo-species biofilms. Archives of oral biology, 145, 105582. https://doi.org/10.1016/j.archoralbio.2022.105582
- Also, still discussing plaque index specifically the 14 day period, the authors should point that 30 days are usually necessary to access this parameter; it is not a general limitation of all the study designed but to this specific results
- When mentioning potencial complications like lipoid pneumonia its worth to mention the children´s group. That is one of the resons why older adults are tested.
_________________________________/_______________________________________
Author Response
Dear reviewer,
We sincerely thank you and the reviewers for the constructive feedback on our manuscript. We have carefully revised the text in accordance with the comments provided. All changes are highlighted in the revised manuscript. Below, we provide a detailed, point-by-point response organized by reviewer. Please see the attachment.
Sincerely,
Assistant Professor Darunee Owittayakul, DDS, MS
Faculty of Dentistry, Chiang Mai University

Reviewer 2 Report
Comments and Suggestions for Authors
The manuscript entitled “A Randomized Trial in Older Adults of a
Flavor-Enhanced Coconut Oil-Based Mouthwash: Clinical Safety,
Antimicrobial Efficacy, and User Satisfaction” has been systematically
conducted with well-structured procedures. However, in order to further
enhance the scientific rigor of the study, I recommend the following
methodological revisions:
Line 151–153: Please provide additional details on how the three groups
(experimental group, positive control group, negative control group) were
formed and allocated.
Line 227: Since the duration of specimen collection from the tongue may
vary among participants, please specify how the specimens were stored and
processed until the analysis of Oral Candida and oral bacteria was
performed.
Line 294: In Table 1, it would strengthen the validity of the study to
include the results of homogeneity tests for demographic and oral health
characteristics across the three groups.
Line 312: In Table 2, when assessing changes in plaque index and oral
moisture levels after 14 days, it may be more insightful to analyze the
interaction effects of each mouthwash. I recommend attempting repeated
measures ANOVA for a more in-depth interpretation.
Line 519–532: The description of study limitations and future research
directions appears somewhat repetitive. I suggest consolidating this section
and additionally clarifying the reasons why the study was limited.
I hope that these suggestions will be helpful, and I wish the authors
continued success in their future research endeavors.
Author Response

(The authors gave the same response as above.)

Reviewer 3 Report
Comments and Suggestions for Authors
Dear Authors,
I have carefully reviewed the submitted manuscript and believe that the paper addresses a topic of major interest, especially in the context of the search for natural alternatives to chlorhexidine for preventive oral care. However, before it can be published, the article requires several significant revisions to improve scientific rigor and readability. In particular, additional methodological details, further clarifications for the interpretation of the results, and a broader contextualization of them in relation to the specialized literature are needed. Here are some recommendations:
Abstract: The study design ("randomized controlled trial") and the allocation method (simple/stratified randomization) should be specified in the Materials and Methods section.
Several values ​​should be added to the results, showing that FCoMW has approximately the same effect as CHX for Candida, but less than CHX for bacteria.
Introduction: Recent statistical data on the prevalence of paraprosthetic oral manifestations in the elderly should be included.
The purpose of the study is not very clearly formulated.
What is the working hypothesis?
Material and method: it should be specified what type of randomized study it is (for example: computerized randomization, block randomization, stratified by sex/age).
Was there blinding (single-blind/double-blind/evaluator-blinded)? This needs to be specified.
The phrase “Candida species were detected by culture from 130 concentrated mouthwash samples, with colony counts not exceeding 400 CF” is not very clear: Did the mouthwash contain Candida colonies?????
What was the total population sample size?
How were subjects recruited (e.g., through clinic advertisements, existing patients, volunteers)?
Exclusion/inclusion criteria need to be detailed.
Did the participants and examiners know the composition of the solutions (was there blinding)?
Additional clarification is needed regarding the reliability of the measurements: for example, for the plaque index, it would be useful to report the Kappa coefficient or the intraclass correlation coefficient (ICC).
For safety parameters, it would be important to mention whether there was a formal protocol for reporting adverse events, including their classification by severity (mild, moderate, severe).
Has the satisfaction questionnaire been validated?
More details about the incubation conditions (aerobic/anaerobic, total duration) should be specified.
Was the identification of Candida colonies done only morphologically or with additional tests (e.g., CHROMagar, PCR, MALDI-TOF)?
For bacteria, were only total CFUs analyzed, or were also target species?
The statistical analysis should be completed with the reporting of confidence intervals (95% CI).
Results: What are the exact p-values, 95% CI, and effect sizes (Cohen’s d, η²)? They should be stated, not just whether the result is significant or not.
Not all data in the tables are found in the text.
Figures are not explicit enough; the values ​​in them should be stated. It is also stated that the results are statistically significant, but no value is given.
Additional exploratory analyses can be performed (e.g., differences between participants with and without dentures, or between those with systemic diseases vs. without), as well as subgroup analyses (denture wearers vs. non-wearers, people with diabetes/HTN vs. healthy, previous mouthwash users vs. non-users).
Do the study results confirm or contradict the working hypothesis?
Discussions: the discussion should be divided into thematic subsections (e.g., Antimicrobial efficacy, Safety and biocompatibility, User acceptability, Limitations and future perspectives).
Do the study results confirm or contradict the literature? What is the significance of these results for clinical practice?
Future directions should be expanded (e.g., multicenter and long-term studies, microbiome sequencing investigations, etc.).
Conclusions: It should be stated why the results are important for dental practice (e.g., for denture wearers, the elderly, patients elderly, patients at risk of oral candidiasis).
Author Response

(The authors gave the same response as above.)

Reviewer 4 Report
Comments and Suggestions for Authors
Dear authors,
Please find below a list of my comments:
Keywords:
- They should be arranged in alphabetical order and correspond to the MeSH database
Introduction
- The null hypothesis is missing after “aim of the study”
(The null hypothesis may possibly be included in the methodology).
Methodology
- The first paragraph of the methodology should always refer to the type of study and its ethics (point 2.2. should be at the beginning of the methodology).
- The date of obtaining the bioethics committee's approval and registration of the study on clinicaltrials.com is missing.
- Inclusion and exclusion criteria are best presented in table form.
- Please refer to the journal's guidelines on how mathematical formulas should be presented.
- As this is a randomized study, there is no information that the study was conducted in accordance with CONSORT guidelines - this information is also missing from the abstract.
Results
- Each table should have all abbreviations used in it expanded below.
Discussion
- At the beginning of the discussion, there should be information on whether the null hypothesis was rejected or accepted. This should be followed by a very brief description of the study results. Only then should we make comparisons with other articles.
- There is no division into subsections (4.1, etc.).
- Please use this source in the discussion:
https://doi.org/10.3390/antibiotics14090876
After taking all comments into account, the article will be suitable for publication.
Best regards
Reviewer
Author Response

(The authors gave the same response as above.)

Round 2
Reviewer 3 Report
Comments and Suggestions for Authors
Dear Authors,
I have reviewed the revised manuscript and found that all submitted comments have been appropriately addressed and clearly integrated into the text. The changes made significantly improve the methodological clarity and scientific value of the paper.
It is recommended, however, to perform a final linguistic check in English, as minor grammatical and formatting errors may persist. It is also useful to make a final detailed revision of the tables and figures (numbering, clarity of legends) before submitting the final version. For example: row 453: table 3 does not contain the data from the text
In conclusion, the article is acceptable for publication after these minor adjustments.
Comments on the Quality of English Language
It is however recommended to carry out a final linguistic check in English, as minor grammatical and wording errors persist.
Author Response
Dear Reviewer,
We sincerely thank you for your thorough evaluation and constructive comments on our revised manuscript. We are truly grateful for your positive feedback acknowledging the improvements in methodological clarity and scientific value.
In accordance with your recommendation, we have performed a comprehensive linguistic revision to ensure grammatical accuracy and consistency throughout the manuscript. In addition, all tables and figures have been carefully rechecked for numbering accuracy, clarity of legends, and correspondence with the text. The issue noted at line 453 has also been corrected, and the table numbering has been revised accordingly (Table 3 revised to Table 4).
We truly appreciate your insightful comments and your kind recommendation for acceptance after minor adjustments. Your valuable feedback has contributed significantly to enhancing the quality and clarity of our manuscript.
Sincerely,
Darunee Owittayakul
(on behalf of all authors)

Reviewer 4 Report
Comments and Suggestions for Authors
Dear authors,
I have no further comments.
Congratulations.
Best regards
Reviewer
Author Response
Dear Reviewer,
Thank you very much for your kind words and for your time and effort in reviewing our manuscript. We greatly appreciate your positive feedback and support.
Best regards,
Darunee Owittayakul